# The Stage-Specific Plasticity of Descending Modulatory Controls in a Rodent Model of Cancer-Induced Bone Pain

**DOI:** 10.3390/cancers12113286

**Published:** 2020-11-06

**Authors:** Mateusz Wojciech Kucharczyk, Diane Derrien, Anthony Henry Dickenson, Kirsty Bannister

**Affiliations:** 1Central Modulation of Pain Group, Wolfson Centre for Age-Related Diseases, King’s College London, London SE1 1UL, UK; kirsty.bannister@kcl.ac.uk; 2Department of Neuroscience, Physiology and Pharmacology, University College London, Gower Street, London WC1E 6BT, UK; diane.derrien.15@ucl.ac.uk (D.D.); anthony.dickenson@ucl.ac.uk (A.H.D.)

**Keywords:** cancer-induced bone pain (CIBP), diffuse noxious inhibitory controls (DNIC), wide dynamic range neurons, neuronal damage, tibial afferents, in vivo electrophysiology, mechanical hypersensitivity

## Abstract

**Simple Summary:**

The mechanisms that underlie pain resulting from metastatic bone disease remain elusive. This translates to a clinical and socioeconomic burden—targeted therapy is not possible, and patients do not receive adequate analgesic relief. The heterogeneous nature of metastatic bone disease complicates matters. Early stage cancers are molecularly very different to their late stage counterparts and so is the pain associated with early stage and advanced tumours. Thus, analgesic approaches should differ according to disease stage. In this article, we demonstrate that a unique form of brain inhibitory control responsible for the modulation of incoming pain signals at the level of the spinal cord changes with the progression of bone tumours. This corresponds with the degree of damage to the primary afferents innervating the cancerous tissue. Plasticity in the modulation of spinal neuronal activity by descending control pathways reveals a novel opportunity for targeting bone cancer pain in a stage-specific manner.

**Abstract:**

Pain resulting from metastatic bone disease is a major unmet clinical need. Studying spinal processing in rodent models of cancer pain is desirable since the percept of pain is influenced in part by modulation at the level of the transmission system in the dorsal horn of the spinal cord. Here, a rodent model of cancer-induced bone pain (CIBP) was generated following syngeneic rat mammary gland adenocarcinoma cell injection in the tibia of male Sprague Dawley rats. Disease progression was classified as “early” or “late” stage according to bone destruction. Even though wakeful CIBP rats showed progressive mechanical hypersensitivity, subsequent in vivo electrophysiological measurement of mechanically evoked deep dorsal horn spinal neuronal responses revealed no change. Rather, a dynamic reorganization of spinal neuronal modulation by descending controls was observed, and this was maladaptive only in the early stage of CIBP. Interestingly, this latter observation corresponded with the degree of damage to the primary afferents innervating the cancerous tissue. Plasticity in the modulation of spinal neuronal activity by descending control pathways reveals a novel opportunity for targeting CIBP in a stage-specific manner. Finally, the data herein have translational potential since the descending control pathways measured are present also in humans.

## 1. Introduction

The mechanisms that underlie pain resulting from bone cancer remain only partially understood. This translates to a clinical and socioeconomic burden—targeted therapy is not possible, and patients do not receive adequate pain relief. The heterogeneous nature of metastatic bone disease complicates matters. Not only does the individual’s pain phenotype depend on genetic, emotional, and sensory factors, but also on the progression of the disease. Early stage cancers are very different to their late stage counterparts and so is the pain associated with early stage and advanced tumours, which may be primary or metastatic.

Animal models of bone cancer are essential for better understanding of the underlying mechanisms that drive this distinct pain state. We have previously shown that injection of syngeneic rat mammary gland adenocarcinoma (MRMT-1) cells in the rat tibia, which manifests a pre-clinical model of cancer-induced bone pain (CIBP) [1], causes increased sensory input to the central nervous system quantified as the recruitment and activation of normally mechanically insensitive nociceptors at day 14 post-injection [2]. As expected, progressive tumour burden also reflects plasticity in central (spinal) events [3,4,5,6]. However, hitherto there is a dearth of data regarding the impact of disease progression on the evoked activity of spinal cord deep dorsal horn wide dynamic range (DDH WDR) neurons. These neurons are of interest since they form a crucial component of spinal neuronal circuits that receive sensory information from the periphery as well as modulation from descending brainstem-origin pathways. In total, their activity reflects global changes in spinal nociceptive processing.

Diffuse noxious inhibitory controls (DNIC) represent a unique top-down modulatory pathway that acts to endogenously reduce the percept of pain via inhibition of DDH WDR neuronal activity [7,8,9]. DNIC and its human counterpart conditioned pain modulation (CPM) is dysfunctional in rodent models of chronic pain and chronic pain patients, respectively [9,10,11,12]. We propose that investigating the functionality of the DNIC pathway in CIBP rats is clinically relevant since pain phenotyping of patients with bone cancer pain has begun to include a measurement of CPM (ClinicalTrials.gov identifier: NCT03908853) using a paradigm previously shown to translate between rodents and humans [13].

It is highly likely that the mechanisms underlying the development of pain in CIBP rats are progressive and therefore representative of distinct molecular changes. We first aim to marry disease progression with behavioural readouts. Thereafter, using an in vivo electrophysiological approach, we will measure the evoked responses of DDH WDR neurons. Is mechanical hypersensitivity in response to threshold and suprathreshold stimulation evidenced behaviourally and electrophysiologically, respectively? Does the top-down modulation of DDH WDR neurons undergo dynamic maladaptive plasticity, and does this occur in a stage-specific manner? Finally, we will investigate whether or not any behavioural and/or spinal changes correlate with a marker of cellular stress in those afferents that innervate the cancer-bearing tibia, in order to link peripheral and central events.

## 2. Results

### 2.1. Bone Destruction in a Rat Model of Cancer-Induced Bone Pain and Early/Late Stage Classification

Following generation of a validated rat CIBP model using syngeneic mammary gland carcinoma cells [1], bone damage caused by cancer growth was evaluated using a high-resolution micro-computer tomography technique (μCT) at two different time points: days seven/eight and days 14/15. Significant damage to the trabecular bone occurred in both stages, while cortical bone was mostly impaired at days 14/15, suggestive of early versus late stage modelling of CIBP as classified in our previous publication [2] (Figure 1A).

### 2.2. CIBP Rats Exhibit Secondary Mechanical Hypersensitivity

Since tumour progression is expected to relate to animal behaviour, rats were monitored up to 14 days post-surgery. While body weight gain remained stable in all groups (Appendix A, [2]), the behavioural data demonstrate that CIBP rats manifest mechanical hypersensitivity. These results correspond to other studies using similar rodent models of CIBP [1,3]. The presence of secondary mechanical sensitization was assessed in sham operated (*n* = 12) and cancer bearing (*n* = 12) rats using the von Frey test. CIBP rats exhibited mechanical allodynia on the side ipsilateral to cancer cell injection in the late stage (from day 14 post-surgery) (two-ay repeated-measures (RM) ANOVA [group]: F _[1, 11]_ = 5.263, *p* = 0.0425, with Bonferroni post-hoc: day 0–day 7: *p* > 0.05, day 14: *p* < 0.0001) (Figure 1B). Both animal groups experienced postsurgical pain in the week following surgery as revealed by a lowered mechanical threshold for von Frey filaments that lasted up until day seven (two-way RM ANOVA [time]: F_[3, 33]_ = 24.05, *p* < 0.0001, Bonferroni post-hoc vs. day 0: sham (day two and day seven): *p* < 0.0001, cancer (days 2–14): *p* < 0.0001). No differences were detected on the contralateral sites in either analysed group (two-way RM ANOVA [group]: F_[1, 11]_ = 0.22, *p* = 0.648, two-way RM ANOVA [time]: F_[3, 33]_ = 3.57, *p* = 0.244, Bonferroni post-hoc vs. day 0: all *p* > 0.05) (Figure 1C).

### 2.3. Deep Dorsal Horn Wide Dynamic Range Neurons Are Not Hyperexcitable in CIBP Rats at Early or Late Stages

The activity of DDH WDR neurons was studied in rats under light isoflurane/N_2_O/O_2_ anaesthesia (slight toe pinch reflex maintained) (Figure 2A). In vivo electrophysiological recordings of DDH WDR neurons were used to study von Frey and brush-evoked firing rates. An example neuronal recording is shown (Figure 2B). Stable baseline neuronal recordings from sham early (SE, *n* = 9), cancer early (CE, *n* = 11), sham late (SL, *n* = 11) and cancer late (CL, *n* = 9) were made. One neuron was studied per animal. Animals with cancer (early or late stage) showed no significant change in the basal firing rate of WDR neurons when compared to sham operated WDR neuronal firing rates. Dynamic brushing of the receptive field (localised typically on the paw) revealed no significant difference between all analysed groups (univariate ANOVA [group]: F_[3, 36]_ = 1.708, *p* = 0.183) (Figure 2C). Analysis of variance revealed no significant changes in the basal von Frey-evoked activity (two-way RM ANOVA [von Frey] F_[3, 108]_ = 217.5, *p* < 0.0001, [group] F_[3, 36]_ = 1.257, *p* = 0.304, Bonferroni post-hoc: all *p* > 0.05) (Figure 2D).

A train of 16 electrical impulses to the receptive field (localised on the hind paw toes ipsilateral to injury) was also applied to verify changes in the basal spinal coding and temporal summation. Example cumulative post-stimulus histograms generated after the delivery of 16 train stimuli from cancer early and late stage rats are shown (Figure 3A). Electrically evoked parameters: Aβ- (F_[3, 36]_ = 0.815, *p* = 0.494), Aδ- (F_[3, 36]_ = 0.543, *p* = 0.656) and C- fibre (F_[3, 36]_ = 3.251, *p* = 0.033, Bonferroni post-hoc: all *p* > 0.05) evoked activity, and input (F_[3, 36]_ = 1.112, *p* = 0.357) and wind-up (F_[3, 36]_ = 2.005, *p* = 0.131) were all unchanged between groups (all: univariate ANOVA [group]) (Figure 3B). Interestingly, post-discharge was significantly reduced in the late stage cancer animals as compared to the early stage ones (F_[3, 36]_ = 3.063, *p* = 0.040, Bonferroni post-hoc: all *p* > 0.05 but CE vs. CL *p* = 0.029) (Figure 3B).

### 2.4. Diffuse Noxious Inhibitory Controls Are Dysfunctional in the Early, but Not Late, Stage of Disease

Previous research has shown that descending modulation of DDH WDR neurons by the inhibitory DNIC pathway is compromised in rodent models of nerve injury and inflammation [4,5]. In the present study, despite the manifestation of behavioural hypersensitivity, the baseline evoked activity of DDH WDR neurons was comparable between sham operated and cancer rats. Therefore, we sought to investigate whether or not descending modulation of those same neurons was dysfunctional during disease progression. DNIC expression was studied in the four aforementioned experimental groups under light isoflurane/N_2_O/O_2_ anaesthesia (slight toe pinch reflex maintained). Terminal electrophysiological recordings of DDH WDR neurons were used to study the von Frey-evoked firing rate changes upon simultaneous ipsilateral application of noxious conditioning stimulus (ear pinch) to evoke DNIC (Figure 4A). One neuron was studied per animal. DNIC were expressed in SE, SL and CL animals, resulting in around 50%, 40% and 30% inhibition of the evoked action potentials to 8 g, 26 g and 60 g von Frey application, respectively (two-way RM ANOVA [von Frey] F_[2, 72]_ = 6.887, *p* = 0.0018) (Figure 4B). Interestingly, DNIC expression was impaired in CE rats (two-way RM ANOVA [group] F_[3, 36]_ = 34.66, *p* < 0.0001, Bonferroni post-hoc: [8 g] CE vs. all *p* < 0.0001, [26 g] CE vs. all *p* < 0.0001, [60 g] CE vs. SE and SL *p* < 0.01, CE vs. CL *p* < 0.0001) (Figure 4B).

In SE animals, DDH WDR neuronal activity was significantly inhibited after ear-pinch when compared to baseline for all von Frey filaments (RM ANOVA: F_[3, 6]_ = 7.669, *p* = 0.018; Bonferroni post-hoc [8 g] *p* = 0.019, [26 g] *p* = 0.003, [60 g] *p* = 0.001) (Figure 4C). In contrast, DDH WDR neuronal responses in CE rats were not inhibited (RM ANOVA: F_[3, 8]_ = 2.606, *p* = 0.124) (Figure 4D). In SL rats, DNIC was expressed (RM ANOVA: F_[3, 8]_ = 13.766, *p* = 0.002; Bonferroni post-hoc [8 g] *p* = 0.004, [26 g] *p* = 0.000047, [60 g] *p* = 0.00106) (Figure 4E), as well as in CL animals (RM ANOVA: F_[3, 6]_ = 37.513, *p* = 0.00028; Bonferroni post-hoc [8 g] *p* = 0.001872, [26 g] *p* = 0.000204, [60 g] *p* = 0.000004) (Figure 4F).

### 2.5. Damage to Primary Afferents Innervating the Cancerous Tissue Was Evident in the Early but Not Late Stage CIBP Rats

Based on our fast blue (FB) tracing, the rat tibia is innervated by 100.7 ± 15.7 lumbar 2–5 dorsal root ganglia (DRG) neurons, which corresponds to 4.47 ± 0.34% of all neurons (based on tubulin-βIII positivity) therein. In health, the majority of those tibia-projecting cells are located within the lumbar 3 (L3) DRG (Appendix A, and [6]).

Damage to afferents innervating cancerous tissue was quantifiable using activating transcription factor 3 (Atf3), a protein induced by cellular stress [7]. Representative micrographs of the L3 DRG ipsilateral to the injury site clearly demonstrated the characteristic nuclear expression pattern of Atf3 in bone and other afferents (Figure 5A). This is especially evident in early stages of our CIBP model (Figure 5A,B). Interestingly, by the late stage, Atf3 positivity normalises in both groups, with almost no occurrence in late-stage sham animals, suggesting full postsurgical recovery (one-way ANOVA [group]: F_[3, 8]_ = 17.29, *p* = 0.0007, Tukey post-hoc: CE vs. SL *p* < 0.001, CE vs. CL *p* < 0.01, SL vs. SE *p* < 0.01) (Figure 5B, Appendix A). Bone afferents are more likely than other afferents to express Atf3 at early disease stages, suggesting higher levels of stress in this population (one-way ANOVA [group]: F_[3, 8]_ = 10.52, *p* = 0.0038, Tukey post-hoc: SE vs. CE *p* < 0.01, CE vs. SL *p* < 0.01, CE vs. CL *p* < 0.05) (Figure 5C). Moreover, there is a visible shift in the expression pattern of Atf3+/FB+ from L3 to lumbar 4 (L4) DRG between early and late CIBP (Appendix A).

Interestingly, no changes were observed in the total number of FB+ neurons between groups when DRG were pooled (one-way ANOVA [group]: F_[3, 8]_ = 3.69, *p* = 0.0062) (Figure 5D); however, there was a shift in tibial innervation from L3 to L4 DRG in cancer-bearing rats (one-way ANOVA [group]: L3 DRG: F_[3, 8]_ = 7.065, *p* = 0.0012, Tukey post-hoc: CE vs. SE *p* < 0.05, CL vs. SL *p* < 0.05, L4 DRG: F_[3, 8]_ = 9.01, *p* = 0.0061, Tukey post-hoc: CE vs. SE *p* < 0.05, CL vs. SL *p* < 0.05, CL vs. SE *p* < 0.01) (Figure 5E,F, Appendix A). Such a shift is suggestive of either degenerative changes in the L3 DRG afferents and/or sprouting of the L4 DRG afferents towards the tumour mass, and/or leakage of the FB tracer outside the bone cavity via cancer-induced perforations of the cortical bone.

## 3. Discussion

In the present study, neuronal activity in the deep dorsal horn of the spinal cord of male Sprague Dawley CIBP rats was investigated. Disease progression was classified as early (day 7–8 post MRMT-1 cells injection) or late (day 14–16) stage according to trabecular and cortical bone destruction. Despite the fact that, upon stimulation of the hind paw, wakeful CIBP rats demonstrated ipsilateral mechanical hyperalgesia in the early stage and mechanical allodynia in the late stage of disease (suggestive of central sensitization), deep dorsal horn wide dynamic range (DDH WDR) neuronal firing upon mechanical stimulation of the same hind paw was unchanged when measured using in vivo electrophysiological techniques. This result was unexpected not only due to the mismatch between behavioural and electrophysiological outcomes but also because spinal cord superficial lamina I neurons were previously consistently described as hyperactive in the CIBP model [3,8,9,10,11]. The divergence in responsiveness of superficial versus DDH WDR neurons could be partly explained based on the anatomy of primary afferents. Lamina I neurons receive direct input from Aδ- and C-fibres as well as silent nociceptors [12]. These afferents predominantly innervate tibiae [2,12,13,14,15]. In contrast, DDH WDR neurons receive direct inputs from large Aβ- and small Aδ-myelinated fibres and indirect polysynaptic inputs from C-fibres from distal dendrites that extend into superficial laminae [16]. The internal spinal circuitry is highly plastic and hugely heterogeneous [17], and the disease state may lead to dysfunctionality in transmission circuits that include inhibitory mechanisms [18], leading to a knock on “adaptive” effect on the evoked activity of DDH WDR neurons, as observed in the present study.

The superficial dorsal horn is the origin of a spino-bulbo-spinal loop and prior investigation of potential alterations in descending modulatory controls was conducted. Rodriguez et al. showed that in CIBP rats both superficial and DDH WDR neurons undergo ongoing facilitatory control (mediated by spinal 5-hydroxytryptamine subtype 3 (5-HT_3_) receptors), suggestive of an enhanced descending serotoninergic drive [10]. Our observation that DDH WDR neurons are not hyperexcitable in the CIBP model could indicate plasticity in ongoing (tonic) inhibitory controls.

Multiple descending inhibitory control pathways exist and activation of one such pathway, diffuse noxious inhibitory controls (DNIC), gives rise to the “pain inhibits pain” phenomenon, whereby application of a noxious stimulus to one part of the body inhibits pain perception in a remote body region. DNIC inhibitory controls are largely driven by α_2_-adrenergic receptor (α_2_-AR)-mediated responses to inhibit the activity of DDH WDR neurons [4]. Notably, during our in vivo spinal electrophysiology studies, we observed a dynamic reorganization of descending inhibitory controls during progression of the bone tumour, quantified as dysfunctional expression of DNIC. Intriguingly, DNIC was abolished only in the early stages of the disease and they were functionally expressed once again in the late stage of CIBP. The dynamic reorganization of spinal neuronal modulation by descending controls corresponds with prior research demonstrating that facilitatory descending controls are altered in CIBP rats [10]. A limitation of our study is that a single sex animal group was used. As with all scientific investigations, including the use of male and female rats (where permissible according to ethical, funding and timeline constraints) in future studies is optimal.

The translational potential of the current study may be considered when discussing the expression of an equivalent naturally occurring analgesic pathway in humans, which is measured using the human psychophysical paradigm conditioned pain modulation (CPM; [19]). Pharmacotherapies that act to enhance the descending inhibitory pathways whose functionality is assessed with CPM psychophysics have already shown promise in enabling chronic pain patients to harness their endogenous pain-relieving mechanisms and thus reduce their pain experience [20]. The presence or absence of CPM is proposed to be a reliable, simple diagnostic measure in terms of personalised pain pharmacotherapies in particular pain types. Recently, several clinical paradigms have been developed for a quantification of the inhibitory impact of CPM [21]. Analogically to DNIC, CPM is pan-modal and it requires test and conditioning stimulus [22] and the conditioning stimulus must be noxious [23]. Recently, a novel approach has utilised two pressure cuffs controlled by a fully automated algometer [24] and there is evidence that a comparable noxious pressure paradigm activates the unique endogenous inhibitory control pathway in rats and humans [25].

Abolished DNIC expression in early stage CIBP rats could serve as an early indicator of the bone cancer pain phenotype development. Through translating our preclinical observations to the clinic, where CPM expression serves as a proxy for functional descending inhibitory controls, one could envisage that abolished expression (as indicated for CPM non-responders, [26]) may be useful as a diagnostic tool that allows targeted pharmacotherapeutic agents to be prescribed. For example, dysfunctional DNIC/CPM has previously been documented in chronic pain animal models and patients, and a beneficial therapeutic effect of pharmacological agents that manipulate noradrenergic transmission has been demonstrated [4,27]. Reassuringly, our recent study outlined that translatable mechanisms underlie endogenous inhibitory pathway expression [25]. In total therefore there is a clinical translational potential in regard to tailored pain pharmacotherapies in patients suffering from (metastatic) bone cancer. Our indication is that stage-specific analgesics, whose mechanism of action accurately reflects the underlying mechanism of central versus peripheral nervous system dysfunction as described in our rodent model of CIBP, could lead to improved analgesic profiles, and therefore should be considered for these patients. Encouragingly, the use of drugs that engage noradrenergic transmission have previously been shown to be effective in preclinical models of CIBP [28,29,30,31]. Interestingly, CIBP is not the only pain state where dynamics in descending controls sub-serving DNIC/CPM have been recorded. CPM is not expressed in patients with cluster headache in the active phase, and yet is restored in remission [32], which suggests that the translatability of our findings regarding stage-specific analgesia that focuses on mechanism-based approaches could extend to multiple chronic pain types.

The sensitization of primary afferent fibres that innervate cancer-bearing bones in the early stage of disease [2,6,33,34,35,36] was linked in the present study with a high expression of cellular stress marker activating transcription factor 3 (Atf3). In the late cancer stage, the activation of Atf3 appeared completely resolved in line with the resolution of DNIC functionality. Multiple mechanisms may contribute to the reduction of Atf3 in the late stages of cancer. For example, it is possible that CIBP afferents are undergoing cell death as a result of tumour invasion and toxic local conditions. Supporting this hypothesis is an observed shift in the expression pattern of Atf3+/FB+ from L3 to L4 DRG between early and late CIBP, further indicating the presence of a degenerative mechanism in L3 tibial afferents, and consecutive sprouting and/or activation of L4 afferents. Indeed, the number of L3 afferents decreases in the cancer groups, as compared to the corresponding sham controls.

Over the past decade, studies of CIBP have revealed that neurons and cancer cells are engaged in bi-directional crosstalk. For instance, cancer causes a reorganization of “normal” anatomy, driving neurons to sprout and more densely innervate the tumour-bearing bone [37,38,39]. This sprouting process was shown to be mediated via tyrosine kinase A (TrkA) receptor activation by nerve growth factor (NGF) released from both cancer and stromal cells [40,41,42]. Conversely, neurons release factors that support tumour growth and vascularization [43,44,45]. This complex dialogue involves numerous mediators and different local cells, including fibroblasts, osteoclasts and newly recruited immune cells [44,46].

Our previously published research has revealed important new information about bone afferent expression patterns, including the fact that they encode mechanical stimuli. Thus, we provided a potential functional mechanism explaining the recruitment of additional afferents from the outside of the tibial cavity that could contribute to the CIBP phenotype. Interestingly, in the presence of the tumour, tibial cavity afferents were not hyperexcitable, a result that was recently confirmed by an independent study on a murine model of CIBP, where femur cavity neurons were not sensitised by the presence of Lewis lung carcinoma tumours after both intraosseous pressure and acid stimulation [33].

In the current study, FB traced tibial cavity neurons were not sensitised in the late stage cancer, which corresponds to the normalised Atf3 level in this group. A decrease in Atf3 levels in the late stage cancer group may, together with the lack of hyperexcitability in this group [2], suggest an ongoing tumour-induced neurodegenerative change in this neuronal population. Supporting this hypothesis, there is a decrease in the total count of FB positive neurons in the L3 DRG in cancer groups as compared to the sham-operated ones, pointing at an ongoing cellular death of this population. A subsequent increase in the number of FB L4 neurons in the cancer groups may result from either neuronal sprouting of the L4 afferents towards the tumour mass (NGF induced: [37,41,47]) and/or leakage of the FB tracer via cancer-induced perforations in the cortical bone or internal bone compartments, resulting in labelling of additional L4 afferents. In fact, since in rats L3 DRG neurons innervate predominantly the medullary cavity and periosteum, and L4 DRG cells innervate epiphysis and since the distal epiphysis and the medullary cavity are not in continuity [15], it is likely that, in our model, cancer-evoked erosion of the medullary cavity towards the epiphysis would result in labelling of additional L4 afferents. It remains to be established whether the long bones CIBP phenotype differs depending on in which bone compartment the tumour grows, i.e., would the CIBP phenotype differ if the tumour invaded the epiphysis to the point where the tumour encompassed the medullary cavity or periosteum?

It is likely that peripheral hyperactivity drives plasticity in central pain controls, a view supported by the aforementioned study demonstrating an increase in descending facilitatory controls (orchestrated via spinal 5HT_3_ receptors) in the CIBP rat dorsal horn neuronal activity [10]. Taking this further, some clusters of primary afferents are modality specific, which superimposes the existence of different synapses on the central (spinal) sites [17,48,49]. Further, the action of descending monoamines (released from the terminals of the descending modulatory pathways investigated) in the cord is rather diffuse, allowing for a broad (inhibitory or excitatory) control of multiple modalities [50,51,52].

CIBP is unique in that the sufferers experience tonic and spontaneous pain as well as the type of mechanically evoked pain studied here. Probing the descending modulatory control of spinal neuronal activity should ultimately include all three types of pain. Background (tonic) pain intensity typically increases with the progression of the disease while spontaneous and movement-evoked types of pain, being mechanoceptive in nature, are difficult to manage in mobile subjects—by definition, they “break-through” the barriers of analgesia [46]. Since they are also unpredictable, it is extremely challenging to find sufficient therapies without adverse effects of high doses of painkillers being continuously administered. Such mechanistic studies were beyond the scope of this paper, but we are aware of these challenges that face preclinical researchers who investigate mechanisms of CIBP.

## 4. Materials and Methods

### 4.1. Cell Lines

Syngeneic rat mammary gland adenocarcinoma cells (MRMT-1, Riken cell bank, Tsukuba, Japan) isolated from female Sprague-Dawley rats, were cultured in Roswell Park Memorial Institute (RPMI)-1640 medium (Invitrogen, Paisley, UK) supplemented with 10% foetal bovine serum (FBS), 1% L-glutamine and 2% penicillin/streptomycin (Invitrogen, Paisley, UK). All cells were incubated at 5% CO_2_ in a humidity-controlled environment (37 °C, 5% CO_2_; Forma Scientific, HongKong, China).

### 4.2. Animals

Male Sprague-Dawley rats (UCL Biological Services, London, UK or Charles-River, Margate, UK) were used for experiments. Animals were group housed on a 12:12 h light–dark cycle. Food and water were available ad libitum. Animal house conditions were strictly controlled, maintaining stable levels of humidity (40–50%) and temperature (22 ± 2 °C). All procedures described were approved by the Home Office and adhered to the Animals (Scientific Procedures) Act 1986. Every effort was made to reduce animal suffering and the number of animals used was in accordance with International Association for Study of Pain (IASP) ethical guidelines [53].

### 4.3. Cancer-Induced Bone Pain Model

On the day of the surgery, MRMT-1 cells were released by brief exposure to 0.1% *w/v* trypsin-ethylenediaminetetraacetic acid (EDTA) and collected by centrifugation in medium for 5 min at 1000 rpm. The pellet was washed with Hanks’ balanced salt solution (HBSS) without calcium, magnesium, or phenol red (Invitrogen, Paisley, UK) and centrifuged for 5 min at 1000 rpm. MRMT1 cells were suspended in HBSS to a final concentration of 300,000 cells/mL and kept on ice until use. Only live cells were counted with the aid of trypan blue (Sigma, NJ, USA) staining. Cell viability after incubation on ice was checked after surgery, and no more that 5–10% of cells were found dead after 4 h of ice-storage.

Sprague-Dawley rats weighing 120–140 g (for late-stage CIBP, 14 days post-surgery) or 180–200 g (for early-stage CIBP, 7 days post-surgery), following complete induction of anaesthesia with isoflurane (induction 5%, maintenance 1.5–2%) in 1 L/min O_2_ and subcutaneous perioperative meloxicam injection (50 μL 2 mg/kg, Metacam^®^, Boehringer Ingelheim, Berkshire, UK) were subjected to the surgical procedure of cancer cell implantation into the right tibiae [1]. Briefly, in aseptic conditions, a small incision was made on a shaved and disinfected area of the tibia’s anterior-medial surface. The tibia was carefully exposed with minimal damage to the surrounding tissue. Using a 0.7 mm dental drill, a hole was made in the bone through which a thin polyethylene tube (I.D. 0.28 mm, O.D. 0.61 mm; Intramedic, Becton Dickinson and Co., Sparks, MD, USA) was inserted 1–1.5 cm into the intramedullary cavity. Using a Hamilton syringe, either 3 × 10^3^ MRMT-1 carcinoma cells in 10 μL HBSS or 10 μL HBSS alone (sham) was injected into the cavity. The tubing was removed, and the hole plugged with bone restorative material (Intermediate restorative material (IRM), Dentsply, Surrey, UK). The wound was irrigated with saline and closed with Vicryl 4-0 absorbable sutures and wound glue (VetaBond 3M, Bracknell, UK). The animals were placed in a thermoregulated recovery box until fully awake.

### 4.4. von Frey Behavioural Testing

Behaviour was assessed 2–4 h before surgery (day 0) and at 2, 7, and 14 days following cancer cells injection. Testing was preceded by a 30 min acclimatisation period. Room conditions used for behavioural testing were strictly controlled, maintaining stable levels of humidity (40–50%) and temperature (22 ± 2 °C). Mechanical hypersensitivity was assessed by application of increment von Frey filaments starting from 0.16 g, up to 26 g–cut off (Touch-test, North Coast Medical Inc., San Jose CA, USA). Each hair was applied 5 times to the plantar surface proximal to the digits of the ipsilateral and contralateral hind paws. Withdrawal responses and whole paw lifts elicited by von Frey hairs were scored as positive remarks. Five subsequent positive responses to the same filament were considered as an overall positive reaction, the force of the filament noted, and further testing with higher force filaments abandoned. Results are presented as a mean ± SEM. The researcher was blinded for quantification.

### 4.5. Spinal Cord In Vivo Electrophysiology

In vivo electrophysiology was performed on animals weighing 250–300 g as previously described [54]. Briefly, after the induction of anaesthesia, a tracheotomy was performed, and the rat was maintained with 1.5% of isoflurane in a gaseous mix of N_2_O (66%) and O_2_ (33%). A laminectomy was performed to expose the L3–L5 segments of the spinal cord. Core body temperature was monitored and maintained at 37 °C by a heating blanket unit with rectal probe. Using a parylene-coated, tungsten electrode (125 μm diameter, 2 MΩ impedance, A-M Systems, Sequim, WA, USA), wide dynamic range neurons in deep laminae IV/V (~650–900 μm from the dorsal surface of the cord) receiving afferent A-fibre and C-fibre input from the hind paw were sought by periodic light tapping of the glabrous surface of the hind paw. Extracellular recordings made from single neurones were visualized on an oscilloscope and discriminated on a spike amplitude and waveform basis. Sampling parameters were set as follows: 30–40k amplification (preamp+amp), band-pass filtering between 1k and 3k Hz and the signal was digitalised at 20 kHz sampling rate. HumBag (Quest Scientific, North Vancouver, BC, Canada) was used to remove low frequency noise (50–60 Hz). Electrical stimulation (NeuroLog system, Digitimer, Welwyn Garden City, UK) was given via two tuberculin needles inserted into the receptive field and a train of 16 stimuli was given (2 ms pulse duration, 0.5 Hz at three times C-fibre threshold: on average 7.8 ± 1.5 mA). Responses evoked by Aβ-, Aδ-, and C-fibres were superimposed and separated according to latency (0–20 ms, 20–90 ms and 90–300 ms, respectively), on the basis that different fibre types propagate action potentials at different conduction velocities. Neuronal responses occurring after the C-fibre latency band of the neuron were classed as post-discharge, a result of repeated stimulation leading to wind-up neuronal hyperexcitability. The “input” (non-potentiated response) and the “wind-up” (potentiated response, evident by increased neuronal excitability to repeated stimulation) were calculated. Input = (action potentials evoked by first pulse at three times C-fibre threshold) × total number of pulses. Wind-up = (total action potentials after 16th train stimulus at three times C-fibre threshold) minus input. Natural mechanical stimuli, including brush and von Frey filaments (2 g, 8 g, 26 g and 60 g), were applied to the receptive field for 10 s per stimulus. For each stimulus, the evoked responses were recorded and quantified as the number of neuronal events counted during the 10 s duration of a given stimulation. Data were captured and analysed by a CED 1401 interface coupled to a PC with Spike 2 software (Cambridge Electronic Design, Cambridge, UK; peristimulus time histogram and rate functions). Stabilization of neuronal responses to the range of electrical and natural stimuli was confirmed with at least three consistent recordings (<10% variation in the action potential) to all measures. Different animals were used to generate the behavioural and electrophysiological data and no animals were excluded from any analysis.

### 4.6. Diffuse Noxious Inhibitory Controls

Diffuse noxious inhibitory controls (DNIC) were induced analogically to previously published methodology [4]. Briefly, extracellular recordings were made from 1 WDR neuron per animal by stimulating the hind paw peripheral receptive field and then repeating in the presence of the ear pinch (conditioning stimulus—DNIC). The number of action potentials fired in 10 s was recorded for each test. Baseline responses were calculated from the mean of 3 trials. Each trial consisted of consecutive responses to 8, 26, and 60 g von Frey filaments applied to the hind paw. This was then followed by consecutive responses to the same mechanical stimuli (8, 26, and 60 g von Frey filaments) in the presence of DNIC. Precisely, DNIC was induced using a noxious ear pinch (15.75 × 2.3 mm Bulldog Serrefine; InterFocus, Linton, United Kingdom) on the ear ipsilateral to the neuronal recording; whilst concurrent to this, the peripheral receptive field was stimulated using the von Frey filaments listed. Diffuse noxious inhibitory control was quantified as an inhibitory effect on neuronal firing during ear pinch. A minimum 30 s non-stimulation recovery period was allowed between each test in the trial. A 10 min non-stimulation recovery period was allowed before the entire process was repeated for control trial number 2 and 3. The procedure was repeated 3 times and averaged only when all neurons met the inclusion criteria of 10% variation in action potential firing for all mechanically evoked neuronal responses.

### 4.7. Immunohistochemistry

For the tracing of intratibial afferents, rats weighing 60–70 g were anaesthetised using isoflurane (1.5–2% in oxygen, Piramal, Northumberland, UK) and the left tibia was injected with 5 μL of 4% fast blue neuronal tracer (Polysciences Inc., Hirschberg an der Bergstraße, Germany). After a 7 day recovery period, animals were randomly divided into two groups—sham and cancer. Then, 7 or 14 days after cancer cells inoculation (for early and late stage, respectively), animals were sacrificed by the overdose of pentobarbital and transcardially perfused with cold phosphate buffer saline (PBS) followed by 4% paraformaldehyde (PFA) in phosphate buffer (pH 7.5). Next, L2–L5 ipsi/contra DRG were collected, post-fixed in 4% PFA, cryo-sectioned and incubated with primary antibodies against Atf3 (rabbit, 1:200, Santa Cruz, (C-19): sc-188, US) and tubulin-βIII (mouse, 1:1000, G712A, Promega, Southampton, UK). Slides were then incubated with the appropriate fluorophore-conjugated secondary antibodies. Representative samples were imaged with a LSM 710 laser-scanning confocal microscope (Zeiss) using 10× (0.3 NA) and 20 × (0.8 NA) dry objectives and analysed with Fiji Win 64. For quantification, samples were imaged with 20× dry objective on a Zeiss Imager Z1 microscope coupled with AxioCam MRm CCD camera. The acquisition of images was made in multidimensional mode and the MosaiX function was used to construct the full view. Furthermore, 3 DRG were imaged per lumbar region. Cell counting was carried out on the Fiji Win 64 utilising the cell counter plugin. For Atf3 analysis, cells were counted as positive only when the cell’s nucleus was stained. The percentage of Atf3 positive cells relative to the total number of neurons (tubβIII) and FB positivity was calculated. On average, 4–20 DRG sections (depending on the DRG size) were imaged for quantification. Moreover, 3 rats per group were used for those experiments and no other procedure was performed on those animals to prevent unspecific activation of Atf3. The researcher was blinded for quantification.

### 4.8. Micro-Computed Tomography of Cancer-Bearing Legs

Rat tibiae, cleared of excess muscle and soft tissue, were placed into a micro-computed tomography scanner (μCT, Skyscan1172) with a Hamamatsu 10 Mp camera. Recording parameters were set as follows: source voltage at 40 kV, source current at 250 μA, rotation step at 0.600 deg, with 2 frames averaging and 0.5 mm aluminium filter. For reconstruction, NRecon software (version: 1.6.10.4) was used. In total, over 500 34 μm thick virtual slices were collected per bone. Cancer growth encompassed an area proximal to the tibial knee head and 114 scan planes covered the majority of the tumour mass (for analysis details see [2]). Representative visualisations were prepared with Fiji with the 3D viewer plugin.

### 4.9. Quantification and Statistical Analysis

Statistical analyses were performed using SPSS v25 (IBM, Armonk, NY, USA). All data plotted in represent mean ± SEM. Throughout the manuscript “n” refers to the number of animals tested. Detailed description of the number of samples analysed and their meanings, together with values obtained from statistical tests, can be found in each figure legend. Symbols denoting statistically significant differences were also explained in each figure legend. Main effects from ANOVAs are expressed as an F-statistic and *p*-value within brackets. Throughout, a *p*-value below 0.05 was considered significant. Behaviour: two-way RM ANOVA with Bonferroni post-hoc test was used to analyse behavioural data for von Frey. Electrophysiology: one-way ANOVA with Bonferroni post-hoc test was used to assess significance for baseline electrical and brush. von Frey responses were assessed with RM ANOVA with the Bonferroni post-hoc. Statistical differences in the neuronal responses observed after ear pinch were determined using a two-way repeated-measures analysis of variance (RM ANOVA) with Bonferroni post-hoc test. One-way ANOVA with Tukey post-hoc performed in the GraphPad Prism was used to analyse immunohistochemical data.

## 5. Conclusions

The overarching aim of the present study was to link previous reports of peripheral sensitization to central (spinal) events. Changes in the peripheral nervous system reflect a notable impact on spinal neuronal responses in the early stage of CIBP, and this is mechanistically linked to dysfunctionality of the descending inhibitory “DNIC” pathway. The data herein provide insight regarding the stage-specific plasticity in central modulatory processes that underlie the pain phenotype in this particular rodent model of CIBP.

## Figures and Tables

**Figure 1 cancers-12-03286-f001:**
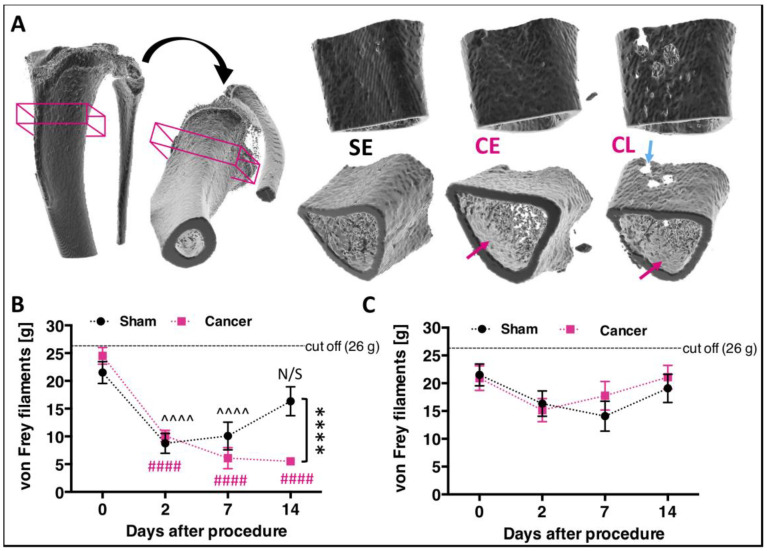
Progressive cancer-evoked bone destruction is reflected in the development of mechanical allodynia. (**A**) Example of microcomputer tomography 3D-rendered rat tibia with corresponding orthogonal projections. The boxed area represents the centre of the cancer-growth zone and constitutes of 114 z-scans taken every 34 μm. Example of the reconstructed cancer growth zone in sham early (SE), cancer early (CE, day 7/8), and cancer late (CL, day 14/15) stages are shown. Blue arrow points at cortical bone lesion, and red arrows point at trabecular bone lesions. See Movie S1 for 360° view. (**B**) von Frey filament-tested mechanical hypersensitivity progression following cancer cells implantation substantially differing from sham-operated control by the late cancer stage (two-way ANOVA with Bonferroni post-hoc: ^^^^ sham, #### cancer vs. day 0, *p* < 0.0001. Day 14 sham vs. cancer: **** *p* < 0.0001, *n* = 12 per group). (**C**) No changes in mechanical sensitivity were observed on the contralateral paw during the whole course of the experiment (two-way repeated-measures (RM) ANOVA with Bonferroni post-hoc, *p* > 0.05, *n* = 12 per group). Results represent mean ± SEM.

**Figure 2 cancers-12-03286-f002:**
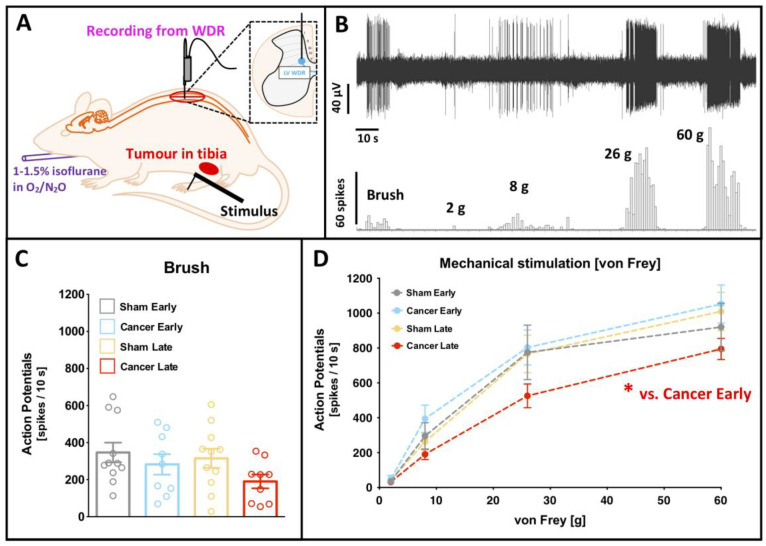
Deep dorsal horn wide dynamic range neurons are less excitable in the late cancer stage to noxious mechanical stimuli. (**A**) Schematic representation of the in vivo electrophysiological experiment. WDR—wide-dynamic range neurons. (**B**) Example of a single cell deep dorsal horn lamina V WDR neuronal responses to dynamic brushing and punctate mechanical stimulation (von Frey filaments) of the receptive field (paw ipsilateral to the cancer) in the late stage bone cancer rat. (**C**) Dynamic brushing-evoked responses of lamina V WDR in early (day 7–8) and late (day 14–16) cancer stage and corresponding sham-operated rats. Each dot represents one animal. One-way ANOVA with Bonferroni post-hoc: *p* > 0.05. (**D**) von Frey-evoked responses of lamina V WDR neurons in early (day 7–8) and late (day 14–16) cancer stage and corresponding sham-operated rats. One-way ANOVA with Bonferroni post-hoc: * *p* < 0.05 cancer early vs. cancer late. All the data represent the mean ± SEM from sham early (*n* = 9), cancer early (*n* = 11), sham late (*n* = 11) and cancer late (*n* = 9).

**Figure 3 cancers-12-03286-f003:**
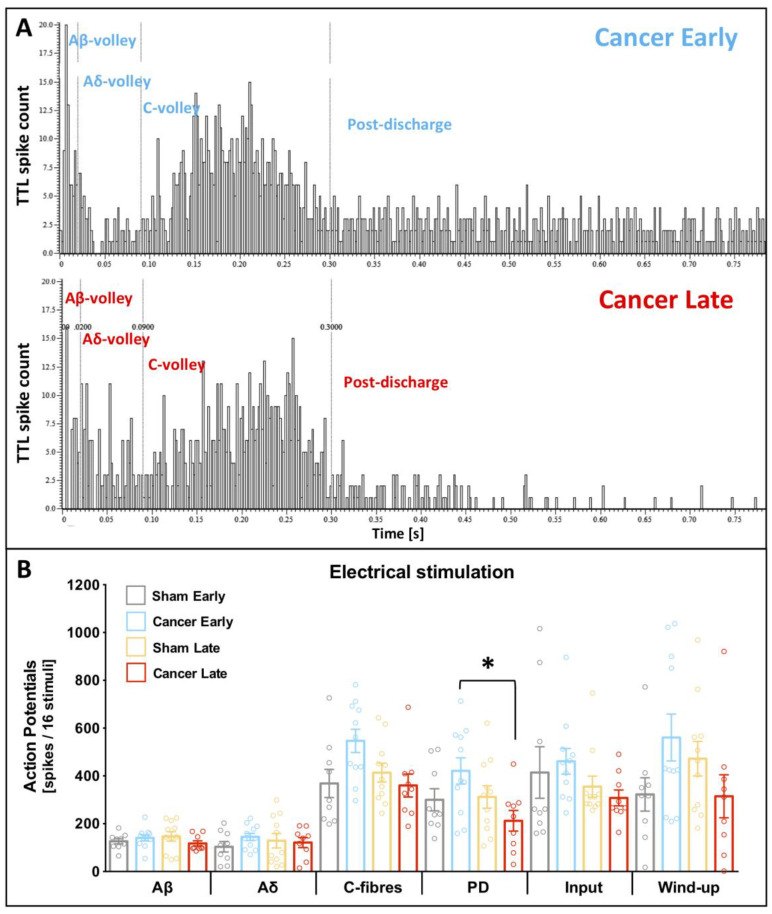
Deep dorsal horn wide dynamic range neurons exhibit shortened post discharge in the late cancer stage. (**A**) Examples of the post-stimulus histograms generated from a single cell deep dorsal horn (lamina V) wide-dynamic range (WDR) neuronal responses recorded in vivo from anaesthetised early (day 7/8, top panel) and late (day 14/15, bottom panel) cancer stage rats. Rats received subcutaneous injection of the current to the peripheral receptive field located on the ipsilateral paw: train of 16 stimuli, 0.5 Hz, 2 ms pulse width, 7.8 ± 1.2 mA mean current. (**B**) Electrically-evoked responses of lamina V WDR neurons in early (day 7–8) and late (day 14–16) cancer stage and corresponding sham-operated rats. Each dot represents one animal. One-way ANOVA with Bonferroni post-hoc: all *p* > 0.05, but post discharge (PD) * *p* < 0.05 cancer early vs. cancer late. All the data represent the mean ± SEM from sham early (*n* = 9), cancer early (*n* = 11), sham late (*n* = 11) and cancer late (*n* = 9).

**Figure 4 cancers-12-03286-f004:**
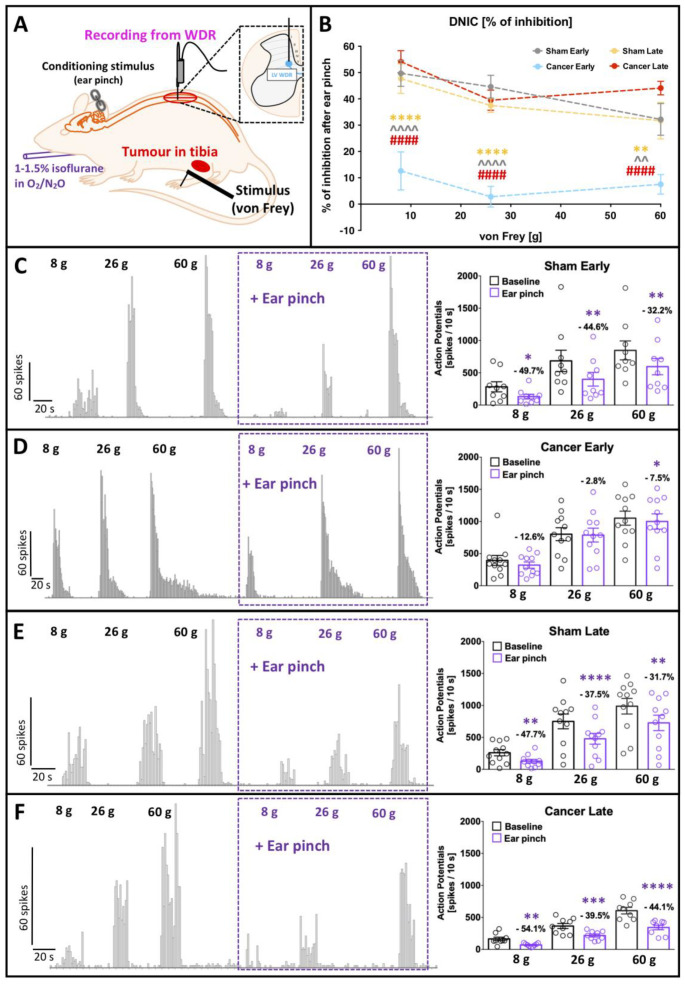
Diffuse noxious inhibitory control expression is compromised in early but not in the late cancer stage. (**A**) Schematic representation of the in vivo electrophysiological experiment. WDR—wide-dynamic range neurons. Activation of diffuse noxious inhibitory controls (DNIC) is quantified as a decrease in von Frey-evoked spinal WDR neuronal firing before (baseline; testing stimulus) and after concomitant application of noxious ear-pinch (DNIC; conditioning stimulus). (**B**) Magnitude of DNIC expression quantified as a percentage of WDR neuron inhibition following ear-pinch application in early (day 7–8) and late (day 14–16) cancer stage and corresponding sham-operated rats. One-way ANOVA with Bonferroni post-hoc: **** *p* < 0.0001 cancer early vs. sham late, ^^ *p* < 0.01, ^^^^ *p* < 0.0001 cancer early vs. sham early, #### *p* < 0.0001 cancer early vs. cancer late. (**C**–**F**) Example of deep WDR neuron responses to increasing bending force of von Frey filaments before and after ear pinch application in sham early (**C**), cancer early (**D**), sham late (**E**) and cancer late (**F**). Individual neuronal responses are quantified in the right panel. Values over each bar represent percentage of change to the respective baseline. Each cell values represent averaged responses from 3 consecutive trials and one cell was recorded per animal (shown as a single dot). Two-way RM ANOVA with Bonferroni post-hoc: * *p* < 0.05, ** *p* < 0.01, *** *p* < 0.001, **** *p* < 0.0001 vs. corresponding baseline. All the data represent the mean ± SEM from sham early (*n* = 9), cancer early (*n* = 11), sham late (*n* = 11) and cancer late (*n* = 9).

**Figure 5 cancers-12-03286-f005:**
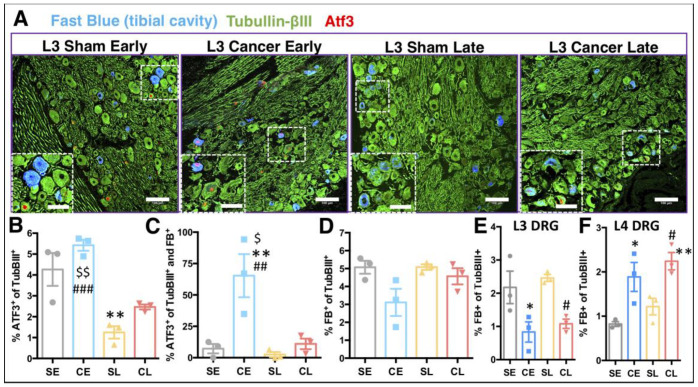
Cancer progression affects bone innervation. (**A**) Representative confocal scans selected from lumbar 3 dorsal root ganglion (DRG) of immunohistochemical analysis of cellular stress factor activating transcription factor 3 (Atf3) and tubulin-βIII protein expression in the fast blue (FB) traced tibial afferents. FB was injected a week before the cancer cells or vehicle (sham) implantation. Main scale bars are 100 μm, and zoomed inclusions’ scale bars are 50 μm. (**B**) Quantification of all Atf3^+^ afferents within ipsilateral L2–5 DRG, analysed as a percentage of all neurons (tubulin-βIII) therein in cancer early (CE, day 7/8) and cancer late (CL, day 14/15) stage groups with corresponding sham groups (early—SE and late—SL). On average 4–20 10 μm sections were counted per DRG. Data represent the mean ± SEM and each dot represent a separate animal (*n* = 3). One-way ANOVA with Tukey post-hoc test. * vs. SE, # vs. SL, $ vs. CL. */#/$ *p* < 0.05, **/##/$$ *p* < 0.01, ### *p* < 0.001. (**C**) Quantification of Atf3+ afferents within ipsilateral L2–5 DRG analysed as a percentage of all FB traced neurons. Analysed as in (**B**). (**D**) Total number of FB traced neurons within ipsilateral L2–5 DRG analysed as a percentage of all neurons (tubulin-βIII) therein. On average, 100.6 ± 15.7 L2–5 DRG neurons innervate tibia. No FB positivity was noticed in the contralateral lumbar DRG (not shown). (**E**) Total number of FB traced neurons in the ipsilateral L3 DRG analysed as a percentage of all neurons (tubulin-βIII) therein. Analysed as in (**B**). (**F**) Total number of FB traced neurons in the ipsilateral L4 DRG analysed as a percentage of all neurons (tubulin-βIII) therein. Analysed as in (**B**). See also Appendix A for detailed analysis.

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
