# Peer review of "The Stage-Specific Plasticity of Descending Modulatory Controls in a Rodent Model of Cancer-Induced Bone Pain"

_cancers, 2020, doi:10.3390/cancers12113286_

Round 1
Reviewer 1 Report
Main Comments:
Authors to explain why they used male rather than female rats for mammary tumour cell-induced bone pain. MRMT-1 breast cancer cells express the estrogen receptor (Barriere et al., Sci Rep. 2019;9:20155, page 2) and hence it would be important for MRMT-1 tumours to be induced in the tibiae of female rather than male rats. In contrast to the claim of ‘rat to human translatability’ of the authors’ main findings, this conclusion is not supported by the findings because male rats were used and breast cancer induced bone pain occurs predominantly in women. Authors please revise the discussion and Abstract sections accordingly as the results from experiments where estrogen-sensitive mammary tumours were induced in the tibiae of male rats, are not directly translatable to female patients with metastatic breast cancer induced bone pain. Using male rats is a major limitation of this work.
Authors please use inclusive language by changing ‘man’ to ‘human’ throughout the manuscript.
Introduction, p2, line 51. What is an ‘infant tumour’?
Add a Supplementary Figure showing the body weight data versus time data for sham- relative to CIBP-groups of rats, as these data are an important indicator of general animal health and so they need to be shown.
Were the sham- and CIBP-groups of animals used to generate the pain behavioural data in Figure 1, the same or different animals to those that were used for in vivo electrophysiology?
P3, Figure 1 legend. Add the word ‘Electronic’ before ‘Von Frey’. This is important as mechanical thresholds in the rat hindpaws determined using manual von Frey filaments are typically in the range, 10-12g in noninjured animals, whereas with electronic von Frey devices, the corresponding baseline von Frey readings are much higher as in this manuscript.
P4, Figure 2 legend and p5-6, Figure 3 legend. The ‘n’ varies between 9 and 11 per group whereas in Figure 1, n=12 per group. What were the reasons for the lower numbers in the groups shown in Figure 2 compared with Figure 1? If data from some animals were excluded, what was the reason for this?
P7, Figure 4 and the associated text. Is DNIC in the ‘cancer late’ group of CIBP-rats potentially underpinned by upregulated endogenous opioidergic signalling? This could be tested by administration of naloxone to inhibit (prevention protocol) or reverse (intervention protocol) DNIC. For comparison, see work by Shenoy et al., Front Pharmacol. 2017;8:442. In Figures 3 and 4 of this paper, there was apparent spontaneous resolution of mechanical hypersensitivity in the hindpaws of breast cancer-induced bone pain in female rats from day 21 to day 28 post intra-tibial injection of rat breast cancer cells and that was maintained for at least 60 days. However, mechanical hypersensitivity was able to be restored in these animals by the systemic administration of naloxone (Figure 8), suggesting that endogenous opioidergic signalling was upregulated in these animals.
P10, Discussion. The paragraph on the ‘translational potential’ and the subsequent paragraph of the Discussion need to be re-written in light of the fact that this study used male rather than female rats which markedly limits the translatable of the findings to women with metastatic breast cancer induced bone pain.
P10 lines 307-308. The authors speculate that CIBP in rats may be sensitive to therapies where noradrenaline re-uptake is inhibited. This has already been shown to be the case by Shenoy et al., Front Pharmacol. 2017;8:442, Figure 7 where the tricyclic antidepressant, amitriptyline, evoked dose-dependent anti-allodynia in the hindpaws of rats with breast cancer induced bone pain, and so it would be appropriate to cite this work in support of your proposal.
P12, Animals, line 400. Why were male and not female rats used given that the MRMT-1 cells express the estrogen receptor? As already stated, the use of male rats markedly limits the translatable of the findings to female patients with metastatic breast cancer induced bone pain.
P13. For the pain behavioural testing in rats, were these measurements performed by testers who were blinded to the treatment group, so as to avoid researcher bias?
P14, lines 468-469. Was DNIC naloxone-sensitive?
P14, sub-section entitled ‘Immunohistochemistry’. Were DRG section images quantified by a researcher blinded to study group to avoid bias?
Minor Points:
P12, line 400. ‘weighing’ not ‘weighting’
P12, LINE 416. ‘Room’ rather than ‘Rooms’
Reviewer 2 Report
This well-performed study aims to dissect the timing of the development of neural plasticity, particularly the descending pain control, in the pathological progression of metastatic bone disease-associated pain. Through an animal model mimicking cancer-induced bone destruction, they provided behavioral, electrophysiological, and immunohistological evidence showing the operation-induced secondary allodynia is attributed to the loss of post-discharge but rather than modified neuronal excitability of wide dynamic range neurons in the dorsal horn. Interestingly, they disclosed altered diffuse noxious inhibitory control that results in sensory innervation to the bone tissue, particularly in the early stage of bone damage, could underlie the modified neural characteristics of wide-dynamic range neurons.
Major comment
The aim of this study is clear, and the experiment design is reasonable. The data is convincing and the manuscript is well presented. Nevertheless, the authors are encouraged to propose the application of such a unique finding in clinical practices. Does early sensory training/electrotherapy is recommended for patients who have evidence of bone metastasis?
Minor comment
All the bar figures are not well-labeled, the words and numbers are not easy to be recognized.
Round 2
Reviewer 1 Report
The authors have suitably addressed the reviewers' comments.